# Disentangled behavioral representations

**Amir Dezfouli**[1][*] **Hassan Ashtiani**[2] **Omar Ghattas**[13]
**Richard Nock**[145] **Peter Dayan**[6][#] **Cheng Soon Ong**[14]

[1]Data61, CSIRO [2]McMaster University [3]University of Chicago
[4]Australian National University [5]University of Sydney [6]Max Planck Institute
Corresponding authors [*]amir.dezfouli@data61.csiro.au [#]dayan@tue.mpg.de

## Abstract

Individual characteristics in human decision-making are often quantified by fitting a parametric cognitive model to subjects' behavior and then studying differences between them in the associated parameter space. However, these models often fit behavior more poorly than recurrent neural networks (RNNs), which are more flexible and make fewer assumptions about the underlying decision-making processes. Unfortunately, the parameter and latent activity spaces of RNNs are generally high-dimensional and uninterpretable, making it hard to use them to study individual differences. Here, we show how to benefit from the flexibility of RNNs while representing individual differences in a low-dimensional and interpretable space. To achieve this, we propose a novel end-to-end learning framework in which an encoder is trained to map the behavior of subjects into a low-dimensional latent space. These low-dimensional representations are used to generate the parameters of individual RNNs corresponding to the decision-making process of each subject. We introduce terms into the loss function that ensure that the latent dimensions are informative and disentangled, i.e., encouraged to have distinct effects on behavior. This allows them to align with separate facets of individual differences. We illustrate the performance of our framework on synthetic data as well as a dataset including the behavior of patients with psychiatric disorders.

## 1 Introduction

There is substantial commonality among humans (and other animals) in the way that they learn from experience in order to make decisions. However, there is often also considerable variability in the choices of different subjects in the same task [Carroll and Maxwell, 1979]. Such variability is rooted in the structure of the underlying processes; for example, subjects can differ in their tendencies to explore new actions [e.g., Frank et al., 2009] or in the weights they give to past experiences [e.g., den Ouden et al., 2013]. If meaningfully disentangled, these factors would crisply characterise the decision-making processes of the subjects, and would provide a low-dimensional latent space that could be used for many other tasks including studying the behavioral heterogeneity of subjects endowed with the same psychiatric labels. However, extracting such representations from behavioral data is challenging, as choices emerge from a complex set of interactions between latent variables and past experiences, making disentanglement difficult.

One promising approach proposed for learning low-dimensional representations of behavioral data is through the use of cognitive modelling [e.g., Navarro et al., 2006, Busemeyer and Stout, 2002]; for example using a reinforcement learning framework [e.g., Daw, 2011]. In this approach, a parametrised computational model is assumed to underlie the decision-making process, and the parameters of this model – such as the tendency to explore and the learning rate – are found by fitting each subject's

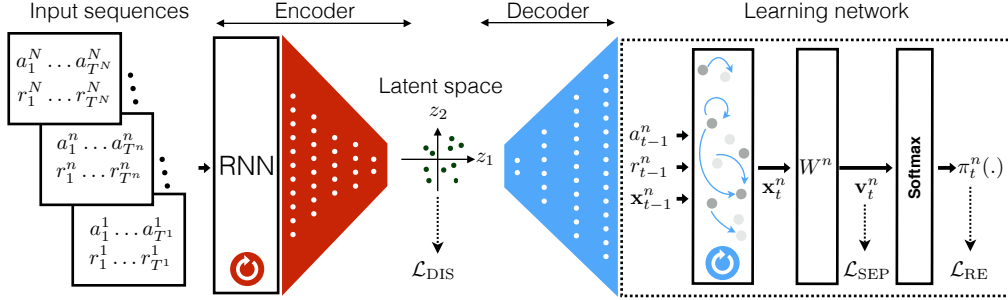

Figure 1: The model comprises an encoder (shown in red) and decoder (blue). The encoder is the composition of an RNN and a series of fully-connected layers. The encoder maps each whole input sequence (in the rectangles on the left) into a point in the latent space (depicted by $(z_1, z_2)$-coordinates in the middle) based on the final state of the RNN. The latent representation for each input sequence is in turn fed into the decoder (shown in blue). The decoder generates the weights of an RNN (called the learning network here) and is shown by the dotted lines on the right side. The learning network is the reconstruction of the closed-loop dynamical process of the subject which generated the corresponding input sequence based on experience. This takes as inputs the previous reward, $r_{t-1}^n$, previous action, $a_{t-1}^n$, and its previous state, $\mathbf{x}_{t-1}^n$, and outputs its next state ($\mathbf{x}_t^n$). $\mathbf{x}_t^n$ is then multiplied by matrix $W^n$ to generate unnormalised probabilities (logits) $\mathbf{v}_t^n$ for taking each action in the next trial, which are converted (through a softmax) to actual probabilities, $\pi_t^n(.)$. The negative-log-likelihoods of the true actions are used to define the reconstruction loss, $\mathcal{L}_{\mathrm{RE}}$, which along with $\mathcal{L}_{\mathrm{DIS}}$ and $\mathcal{L}_{\mathrm{SEP}}$ are used to train the model. The $\mathcal{L}_{\mathrm{DIS}}$ term induces disentanglement at the group level, and $\mathcal{L}_{\mathrm{SEP}}$ separates the effects of each dimension of the latent space on the output of the learning network.

choices. Their individual parameters are treated as the latent representations of each subject. This approach has been successful at identifying differences between subjects in various conditions [e.g., Yechiam et al., 2005]; however, it is constrained by its reliance on the availability of a suitable parametric model that can capture behavior and behavioral variability across subjects. In practice, this often leads to manually designing and comparing a limited number of alternative models which may not fit the behavioral data closely.

An alternative class of computational models of human decision-making involves Recurrent Neural Networks (RNNs) [e.g., Dezfouli et al., 2019, 2018, Yang et al., 2019]. RNNs can model a wide range of dynamical systems [Siegelmann and Sontag, 1995] including human decision-making processes, and make fewer restrictive assumptions about the underlying dynamical processes. However, unlike cognitive models, RNNs typically: (i) have large numbers of parameters, which (ii) are hard to interpret. This renders RNNs impractical for studying and modelling individual differences.

Here we develop a novel approach which benefits from the flexibility of RNNs, while representing individual differences in a low-dimensional and interpretable space. For the former, we use an autoencoder framework [Rumelhart et al., 1985, Tolstikhin et al., 2017] in which we take the behaviors of a set of subjects as input and automatically build a low-dimensional latent space which quantifies aspects of individual differences along different latent dimensions (Figure 1). As in an hyper-networks [Ha et al., 2016, Karaletsos et al., 2018], the coordinates of each subject within this latent space are then used to generate the parameters of an RNN which models the decision-making processes of that subject. To address interpretability, we introduce a novel contribution to the autoencoder's loss function which encourages the different dimensions of the latent space to have separate effects on predicted behavior. This allows them to be interpreted independently. We show that this model is able to learn and extract low-dimensional representations from synthetically-generated behavioral data in which we know the ground truth, and we then apply it to experimental data.

## 2   The model

**Data.**   The data $\mathcal{D} \equiv \{\mathcal{D}^n\}_{n=1}^N$ comprise $N$ input sequences, in which each sequence $n \in \{1 \dots N\}$ consists of the choices of a subject on a sequential decision-making task. In input sequence $n \in$

$\{1, \ldots, N\}$ the subject performs $T^n$ trials, $\mathcal{D}^n \equiv \{(a_t^n, r_t^n)\}_{t=1}^{T^n}$, with action $a_t^n \in C$ on trial $t$ chosen from a set $C \equiv \{C_k\}_{k=1}^K$, and reward $r_t^n \in \Re$.

**Encoder and decoder.** We treat decision-making as a (partly stochastic) dynamical system that maps past experiences as input into outputs in the form of actions. As such, the dataset $\mathcal{D}$ may be considered to contain samples from the output of potentially $N$ different dynamical systems. The aim is then to turn each sequence $\mathcal{D}^n$ into a vector, $\mathbf{z}^n$, in a latent space, in such a way that $\mathbf{z}^n$ captures the characteristic properties of the corresponding dynamical system (whilst avoiding over-fitting the particular choices). In this respect, the task is to find a low-dimensional representation for each of the dynamical systems in an unsupervised manner. We take an *autoencoder*-inspired model to achieve this, in which an encoder network is trained to process an entire input sequence into a vector in the latent space. A decoder network then takes as input this vector and recovers an approximation to the original dynamical system. Along with additional factors that we describe below, the model is trained by minimising a reconstruction loss which measures how well the generated dynamical system can predict the observed sequence of actions. The latent representation is thus supposed to capture the "essence" of the input sequence. This is because the latent space is low-dimensional compared to the original sequence and acts as an information bottleneck; as such the encoder has to learn to encode the most informative aspects of each sequence.

The model architecture is presented in Figure 1. The first part is an encoder RNN (shown in red), and is responsible for extracting the characteristic properties of the input sequence. It takes the *whole* sequence as input and outputs its terminal state. This state is mapped into the latent space through a series of fully-connected feed-forward layers. The encoder is:

$$\mathbf{z}_{M \times 1}^n \equiv \mathrm{enc}(a_{1 \ldots T}^n, r_{1 \ldots T}^n; \Theta_{\mathrm{enc}}), \quad n = 1 \ldots N, \tag{1}$$

in which $\Theta_{\mathrm{enc}}$ are the weights of the RNN and feed-forward layers, and $M$ is the dimensionality of the latent space (see Supplementary Material for the details of the architecture).

The second part of the model is a feed-forward decoder network (shown in blue) with weights $\Theta_{\mathrm{dec}}$, which takes the latent representation as input, and outputs a vector $\Phi^n$,

$$\Phi^n \equiv \mathrm{dec}(\mathbf{z}^n; \Theta_{\mathrm{dec}}). \tag{2}$$

As in a hyper-network, vector $\Phi^n$ contains the weights of a second RNN called the *learning network*, itself inspired by [e.g., Dezfouli et al., 2019, 2018], and described below. The learning network apes the process of decision-making, taking past actions and rewards as input, and returning predictions of the probability of the next action. Making $\Phi^n$ such that the learning network reconstructs the original sequence is what forces $\mathbf{z}^n$ to encode the characteristics of each subject.

**Learning network.** The learning network is based on the Gated Recurrent Unit architecture [GRU; Cho et al., 2014] with $N_c$ cells. This realizes a function $f^n$, which at time-step $t$ maps the previous state $\mathbf{x}_{t-1}^n$, action $a_{t-1}^n$ and reward $r_{t-1}^n$, into a next state,

$$\mathbf{x}_t^n \equiv f^n(a_{t-1}^n, r_{t-1}^n, \mathbf{x}_{t-1}^n; \Phi^n), \tag{3}$$

with predictions for the probabilities of actions arising from a weight matrix $W^n \in \Re^{K \times N_c}$

$$\mathbf{v}_t^n \equiv W^n \mathbf{x}_t^n, \quad \pi_t^n(a_t \equiv C_i; a_{1 \ldots t-1}^n, r_{1 \ldots t-1}^n) = \frac{e^{\mathbf{v}_t^n[i]}}{\sum_{k=1 \ldots K} e^{\mathbf{v}_t^n[k]}}, \tag{4}$$

where $\mathbf{v}_t^n$ represent the logit scores for each action (unnormalised probabilities), and $\pi_t(.)$ are the probabilities of taking each action at time $t$. $\mathbf{v}_t^n[i]$ represents the $i$th element of $\mathbf{v}_t^n$. For $N_c$ GRU cells and $K$ actions, function $f^n$ requires $J = 3N_c^2 + 3KN_c + 6N_c$ parameters. $\Phi^n$ consists of these plus the $KN_c$ parameters of $W^n$. Note that this RNN, which models humans decision-making processes, should not be confused with the RNN in the encoder, which extracts and maps each input sequence into the latent space.

**Summary**. The encoder network takes an input sequence and maps it into its latent representation. The decoder network then takes the latent representation and generates the weights of an RNN that is able to predict the actions taken in the input sequence. The next section describes in detail how the network weights $\Theta_{\mathrm{enc}}$ and $\Theta_{\mathrm{dec}}$ are learned end-to-end.

## 3 Training objective

The training loss function has three components: (i) a reconstruction loss which penalizes discrepancies between the predicted and actual input sequence, (ii) a group-level disentanglement loss which encourages sequences to spread independently across the dimensions of the input sequence, (iii) a separation loss which favors dimensions of the latent space that have separate effects on the behavior generated by the learning networks.

**Reconstruction loss.** Each sequence $\mathcal{D}^n$ in $\mathcal{D}$ is passed through the encoder to obtain the corresponding latent representation. The latent representation is then passed through the decoder to generate the weight vector $\Phi^n$ of the learning network. We then assess how well the $n$-th learning network can predict the actions taken in $\mathcal{D}^n$. The total reconstruction loss is based on the negative-log-likelihood:

$$\mathcal{L}_{\text{RE}} = -\frac{1}{N} \sum_{n=1}^{N} \sum_{t=1}^{T^n} \log \pi_t^n(a_t^n; a_{1...t-1}^n, r_{1...t-1}^n). \tag{5}$$

**Disentanglement loss.** The second term in the loss function favors disentangled representations by seeking to ensure that each dimension of the latent space corresponds to an independent factor of contribution in the variation across input sequences. This is achieved by maximising the similarity between the empirical encoding distribution (i.e., $\{\mathbf{z}^n\}$) and a prior distribution $p(\mathbf{z})$, taken to be isotropic Gaussian (see the Supplementary Material for details). Maximising this similarity encourages the dimensions of the latent representation to be independent across input sequences. Define $\hat{q}(\mathbf{z})$ as the empirical distribution of $\mathbf{z}^n$, and $g(\mathbf{z})$ as a Gaussian with the same mean and covariance as that of $\hat{q}(\mathbf{z})$. We combine two popular measures of the dissimilarity between the encoding distribution and $p(\mathbf{z})$:

$$\mathcal{L}_{\text{DIS}} = \lambda_1 \text{MMD}(\hat{q}(\mathbf{z}), p(\mathbf{z})) + \text{KL}(g(\mathbf{z}) \| p(\mathbf{z})), \tag{6}$$

where MMD is the Maximum Mean Discrepancy [Tolstikhin et al., 2017] and KL is the Kullback-Leibler divergence (the Gaussian approximation makes the KL computations tractable and robust) and $\lambda_1$ is a hyper-parameter. We expected the MMD term to suffice; however, we found that combining the two yielded better results.

**Separation loss.** Crudely speaking, the disentanglement loss focuses on regularising the encoder; however, to make the latent representation behaviorally interpretable, it is also necessary to regularize the decoder. Gaining insight into the effect of one dimension on the behavioral predictions of the learning network is hard if this is affected by the other dimensions. Therefore, we introduce an additional separation loss designed to discourage interactions.

For simplicity, assume the decision-making task involves only two actions, $C_1$ and $C_2$, and that the latent space has two dimensions, $z_1$ and $z_2$ ($M = 2$; see Supplementary Material for the general case). As noted before $\mathbf{v}_t^n(C_1)$ and $\mathbf{v}_t^n(C_2)$ are the logits corresponding to the probability of taking action $C_1$ and $C_2$ at trial $t$ for input sequence $n$. Denote by $u_t^n$ the relative logits of the actions, $u_t^n = \mathbf{v}_t^n(C_1) - \mathbf{v}_t^n(C_2)$. The amount that $u_t^n$ changes by changing the first dimension, $z_1$ is

$$\left| \frac{\partial u_t^n}{\partial z_1} \right|. \tag{7}$$

Ideally, the effect of changing $z_1$ on behavior will be independent from the effect of $z_2$ on behavior, which will allow us later to interpret $z_1$ independently from $z_2$. We capture the amount of interaction (inseparability) between the effect of $z_1$ and $z_2$ on changing $u_t^n$ as

$$d_t^n = \left| \frac{\partial^2 u_t^n}{\partial z_1 \partial z_2} \right|. \tag{8}$$

This would be zero if the relative logit comprises *additively separable functions*, i.e., $u_t^n = g_1(z_1) + g_2(z_2)$ for two functions $g_1$ and $g_2$. Even without this, minimizing $d_t^n$ can help make each dimension have a separate effect on behavior. We therefore consider the following loss term,

$$\hat{\mathcal{L}}_{\text{SEP}} = \frac{1}{N} \sum_{n=1}^{N} \sum_{t=1}^{T^n} \left| \frac{\partial^2 u_t^n}{\partial z_1 \partial z_2} \right|. \tag{9}$$

The calculation of the above term is computationally intensive as it requires computing the second-order derivative for each time step and sequence pair ($\mathcal{O}(NT^n)$ second-order derivative calculations). Instead, we use the following approximation,

$$\hat{\mathcal{L}}_{\text{SEP}} \approx \mathcal{L}_{\text{SEP}} = \frac{1}{N}\sum_{n=1}^{N}\left|\sum_{t=1}^{T^n}\frac{\partial^2 u_t^n}{\partial z_1 \partial z_2}\right| = \frac{1}{N}\sum_{n=1}^{N}\left|\frac{\partial^2}{\partial z_1 \partial z_2}\sum_{t=1}^{T^n}u_t^n\right|, \qquad (10)$$

which can be calculated more efficiently ($\mathcal{O}(N)$ second-order derivative calculations)[1]. We note that the loss is defined using logits instead of the probabilities, since probability predictions are bounded and cannot be separated as $\pi_t^n(C_1) - \pi_t^n(C_2) = g_1(z_1) + g_2(z_2)$ in the general case.

We now define the combined loss function as follows,

$$\mathcal{L} = \mathcal{L}_{\text{RE}} + \lambda_2 \mathcal{L}_{\text{DIS}} + \lambda_3 \mathcal{L}_{\text{SEP}}, \qquad (11)$$

in which $\lambda_2$ and $\lambda_3$ are hyper-parameter.

The model parameters $\Theta_{\text{enc}}$ and $\Theta_{\text{dec}}$ were trained based on the above objective function and using gradient descent optimisation method [Kingma and Ba, 2014]. See Supplementary Material for details.

## 4 Results

**SYNTHETIC data.** To illustrate that our method can learn the underlying ground truth dynamical system, we generated $Q-$learning agents [Watkins, 1989] with various parameter values and simulated their behaviour on a bandit task involving two stochastically-rewarded actions, $C_1$ and $C_2$. The actions of the agents and the rewards they received comprise the dataset $\mathcal{D}$. The $Q-$learning agent was specified by two parameters, values for which were drawn randomly (see Supplementary Material for more details). One parameter is the inverse temperature or reward sensitivity, $\beta$, which determines the propensity to explore, equivalently controlling the impact of receiving a reward from an action on repeating that action in future trials. The other parameter is $\kappa$, which determines the tendency of an agent to repeat the last taken action in the next trial irrespective of whether it was rewarded [e.g., Lau and Glimcher, 2005]. Values $\kappa > 0$ favor repeating an action in next trial (perseveration) and values $\kappa < 0$ favor switching between the actions. We generated $N = 1500$ agents (saving 30% for testing). The test data was used for determining the optimal number of training iterations (early stopping). Each agent selected 150 actions ($T^n = 150$).

We trained our model using the data (see Figure S3 for the trajectory of the loss function and Figure S4 for the distribution of the latent representations). As shown in Figure 2(a), and as intended, these representations turned out to have an interpretable relationship with the parameters that actually determined behavior: the exploration parameter $\beta$ is mainly related to $z_2$ and the perseveration parameter $\kappa$ to $z_1$. This outcome arose because of the $\mathcal{L}_{\text{SEP}}$ term in the loss function. A similar graph *before* introducing the $\mathcal{L}_{\text{SEP}}$ is shown in Figure S1(a), which shows that without this term, $z_1$ and $z_2$ have mixed relationships with $\kappa$ and $\beta$.

We then sought to interpret each dimension of the latent space in behavioral terms. To do this, we used the decoder to generate learning networks corresponding to different values of the latent representations and interpreted these dimensions by analysing the behaviour of the simulated networks on a fixed input sequence. In the fixed input sequence, the agent always performs action $C_1$ (see Figure S5 for the simulations using both $C_1$ and $C_2$ as the previous action), and receives just one non-zero reward at the trial marked by the red vertical line. Using the same sequence for all networks requires running simulations off-policy (i.e., the network predicts the next choice, but does not execute it). This setting provides the same input to the model in all conditions, allowing us to diagnose exactly what affects the output of the model. The simulations are shown in Figure 2(b). Each panel in the figure shows the simulation of the generated network for 10 trials.

In Figure 2(b)-left panel, $z_2$ is fixed at zero as $z_1$ is allowed to vary. As the figure shows, by changing the $z_1$ dimension, the probability of taking the same action (in this case $C_1$) in the next trial is affected, i.e., high values of $z_1$ are associated with the high probability of perseveration, and in the low values of $z_1$ there is a high chance of switching to the other action in the next trial. This is consistent with

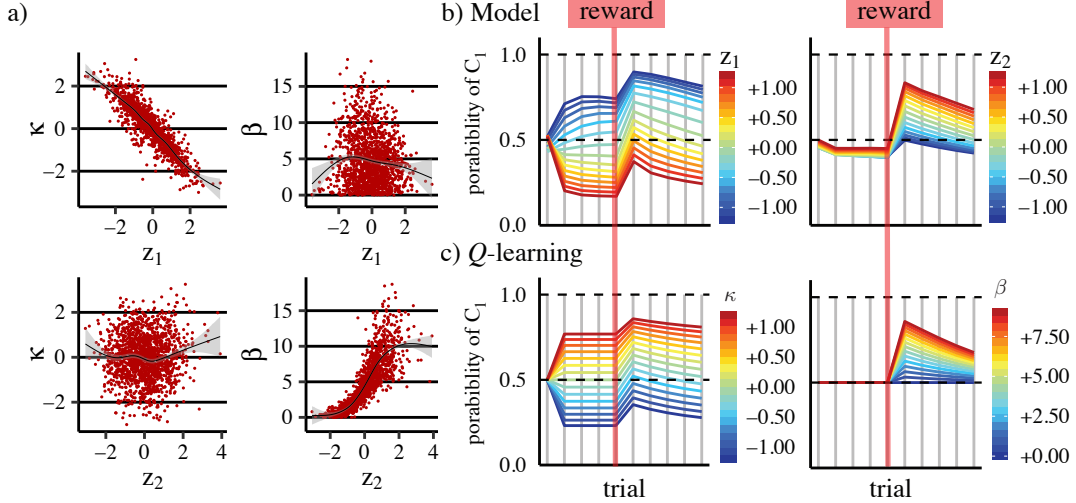

Figure 2: SYNTHETIC data. (a) Relationship between the dimensions of the latent representations ($z_1$, $z_2$) and the parameters used to generate the data ($\kappa$ and $\beta$). $z_1$ captures the perseveration parameter ($\kappa$) and $z_2$ captures the rate of exploration ($\beta$). The smoothed black lines were calculated using method 'gam' in R [Wood, 2011] and the shaded area represents confidence intervals. (b) Off-policy simulations of the model for different values of $z_1$ (left-panel; $z_2 = 0$) and $z_2$ (right-panel; $z_1 = 0$). The plots show the probability of selecting $C_1$ in each trial when $C_1$ had actually been chosen on all the previous trials. A single reward is provided, shown by the vertical red line. (c) Model simulations similar to the ones in panel (b) but using the actual $Q-$learning model. In the left panel $\beta = 3$ and in the right panel $\kappa = 0$.

the correspondence between $z_1$ and the perseveration parameter $\kappa$. Note that after receiving the reward, the probability of taking $C_1$ increases (since $C_1$ was the previous action and it was rewarded), however, the size of the increment is not affected by changes in $z_1$, i.e., $z_1$ is controlling perseveration but not sensitivity of behaviour to reward.

In contrast in Figure 2(b)-right panel, $z_1$ is fixed at zero as $z_2$ is in turn allowed to vary. As the panels show, by changing $z_2$ the probability of repeating an action is not affected, but the sensitivity of action probabilities to the reward is affected. That is, at the higher values of $z_2$ there is a high probability of repeating the rewarded action ($C_1$ in this case). Therefore, $z_1$ and $z_2$ have separate effects on behaviour corresponding to the perseveration and reward sensitivity. This separation is a consequence of introducing the $\mathcal{L}_{\text{SEP}}$ term. In Figure S1(b) we show the same simulations but without introducing the $\mathcal{L}_{\text{SEP}}$ into the loss function, which shows that the effects of $z_1$ and $z_2$ on behaviour are *not* separated (see Figure S2 for how the behaviour of model changes during training). Figure 2(c) shows the same simulations as panel (b) but using the original $Q$-learning model which was used to generate the data (for different values of $\beta$ and $\kappa$). As the figure shows the model closely reflects the behaviour of $Q-$learning model, as expected.

**BD dataset.** This dataset [Dezfouli et al., 2019] comprises behavioural data from $34$ patients with depression, $33$ with bipolar disorder and $34$ matched healthy controls. As in the synthetic data above, subjects performed a bandit task with two stochastically-rewarded actions ($C_1$ and $C_2$) (Figure 3(a)). Each subject completed the task $12$ times using different reward probabilities for each action. The dataset thus contains $N = 12$ (sequences) $\times 101$ (participants) $= 1212$, which we used for training the model. Out of the $12$ sequences of each subject, $8$ were used for training and $4$ for testing to determine the optimal number of training iterations (see Figure S7 for the training curves and Supplementary Material for more details).

We considered a two-dimensional latent space $\mathbf{z} = \{z_1, z_2\}$; the resulting coordinates for all sequences are shown in Figure 3(b). We made two predictions about $\mathbf{z}$: first, we expected that the latent representations for the sequences for a single subject should be mutually similar as they come from the same decision-making system. We therefore compared the mean pairwise distances separating

the latent representations within and between subjects (see Supplementary Material for details). Figure 4(a) shows that this prediction is indeed correct ($p < 0.001$ using Wilcoxon rank sum test).

Second, we expected that the decision-making differences between the three groups to be reflected in their latent representations. Figure 4(b) shows the mean latent representations for the subjects organized by group. Although the groups differ along the $z_2$ dimension ($p < 0.001$ comparing bipolar and healthy groups along $z_2$ and using independent t-test, and $p < 0.05$ comparing depression and healthy groups); the $z_1$ dimension is evidently capturing variations that are unrelated to the existing diagnostic categories ($p > 0.1$).

These results also highlight the substantial heterogeneity of the population. Some bipolar patients with high $z_2$ are apparently more behaviorally similar to average depressed patients or healthy controls than to the average bipolar patients. We would not have been able to extract this information by fitting a single RNN to each whole group (as was done in previous work).

We then followed the same scheme as for the synthetic data to provide an interpretation for the dimensions of latent space. The results of off-policy simulations are shown in Figure 4(c) (see Figure S6 for simulations using both actions). First, both left and right panels show that, after receiving the reward from an action, subjects show a tendency to *switch* to the other action. This is inconsistent with predictions of conventional $Q-$learning, but was also found to be evident in model agnostic analyses [Dezfouli et al., 2019] (where it is also reconciled with gaining rather than losing reward). The current model is able to capture this since it uses RNNs for representing decision-making processes, which do not make such an assumption.

Second, as Figure 4(c)-left shows, the $z_1$ dimension – which is not different between the groups – is mostly related to reward sensitivity, as it controls (albeit rather weakly) the action probabilities after receiving the reward. On the other hand, the $z_2$ dimension – which is significantly different between the groups – is more involved in perseveration/switching behaviour between the actions, i.e., it controls the probability of staying on the same action. For low $z_2$, the probability of repeating the previous taken (in this case $C_1$) is below $0.5$, implying that the subjects in this part of the latent space tend to switch. In order to confirm this, we simulated the model on-policy – in which the actions are selected by the model – for different values of $z_1$ and $z_2$. As the results in Figure 4(d) show, for low $z_2$, the model indeed oscillates between the two actions. The $z_2$ dimension is significantly lower in the bipolar group than healthy controls, which is consistent with the previous report indicating an oscillatory behavioural characteristic in this group [Dezfouli et al., 2019].

Finally, panel (c)-right shows that reward sensitivity and perseveration are not completely independent, i.e., when $z_2$ is low, favoring switching, the effect of reward on probabilities is also more significant, implying that these two traits covary.

In summary, the model was able capture behavioural properties which are not consistent with cognitive models such as $Q-$learning. It was also able to capture individual differences which could not be extracted by fitting a single RNN to the whole group.

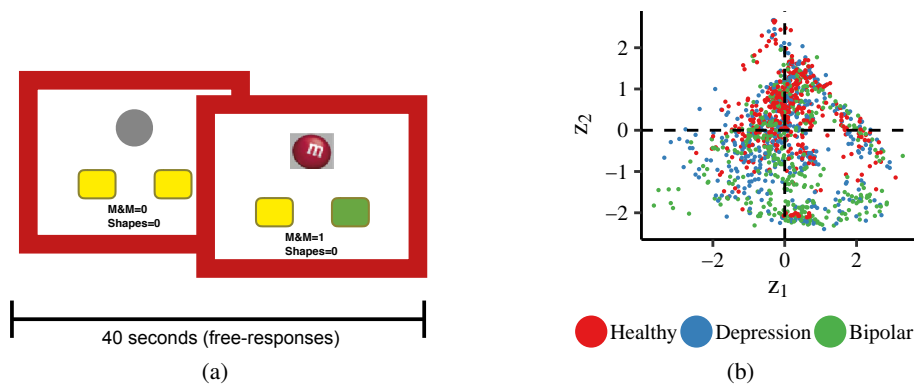

(a)
(b)

Figure 3: BD dataset. (a) The decision-making task. On each trial, subjects pressed a left ($C_1$) or right ($C_2$) key and had a chance of getting a reward (M&M chocolate or BBQ shapes). The task lasted for 40 seconds and the responses were self-paced. Each participant completed the task 12 times with a 12-second delay between them and different probabilities of reward. (b) Distribution of **z** values for each group. Each dot represents an input sequence.

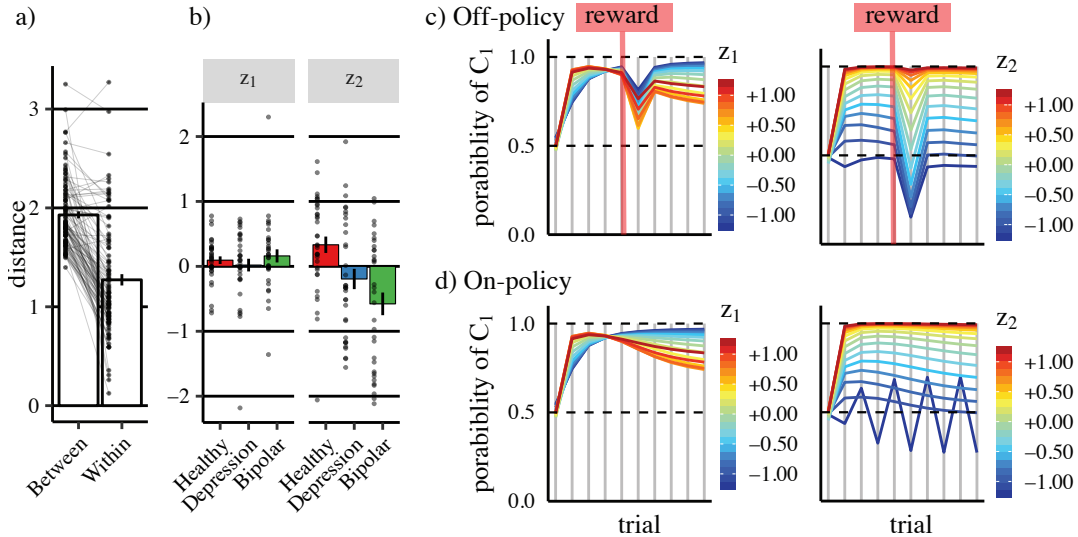

Figure 4: BD dataset. (a) The distances between the latent representations for single subjects (Within) and between the subjects (Between). Each dot represents a subject; the bars show the means and the error-bars, 1SEM. (b) The mean of the latent representations for each subject across the dimensions of the latent space. Each dot represents a subject and bars and error-bars show means and 1SEM. (c) Off-policy simulations of the model for different values of $z_1$ (left-panel; $z_2 = 0$) and $z_2$ (right-panel; $z_1 = 0$). The plots show the probability of selecting $C_1$ in each trial when $C_1$ had actually been chosen on all the previous trials. A single reward is provided, shown by the vertical red line. (d) On-policy simulations of the model. The actions are selected by the model based on which action has the higher probability of being taken (the first action was set to $C_1$).

## 5 Related work

There is a wealth of work using RNNs as models of decision-making, for unsupervised dimension reduction of dynamical systems, and for sequence-to-sequence mappings. For the first, a key focus has been learning-to-learn [e.g., Wang et al., 2016, Song et al., 2017] – i.e., creating RNNs that can themselves learn to solve a battery of tasks. However, these generative models have not been coupled with recognition, for instance to capture individual differences. Techniques based on autoencoders have explored low-dimensional latent spaces for modelling neural activity trajectories in the brain [e.g., Pandarinath et al., 2018]. However, like influential sequence-sequence models for translation [Bowman et al., 2015], these focus on building an open loop account for the *state* of the brain within a trajectory, or the *state* characterizing a particular input sentence, whereas we focus on the *trait* characteristic of a closed-loop controller that has to interact with an external environment itself [see also Fox et al., 2011, Johnson et al., 2016, as examples of other approaches]. Our combination of an autoencoder and hyper-network [Ha et al., 2016, Karaletsos et al., 2018] is, to our knowledge, novel, and might find other applications in the analysis of other dynamical systems such as time-series data.

## 6 Conclusion and discussion

We proposed a flexible autoencoder-based framework for modelling individual differences in decision-making tasks. The autoencoder maps the sequence of actions and rewards taken and received by a subject to a position in a low-dimensional latent space which is decoded to determine the weights of a 'learning network' RNN that characterizes how that subject behaves as a plastic decision-maker. The latent space was disentangled by adding a specific component to the loss function.

The model can be extended in various directions. One is to pin down (part of) the learning network as a conventional, parameterized, RL agent, to facilitate further interpretation of the latent dimensions. Another is to include tasks with non-trivial (Markovian or non-Markovian) state information partially signalled by stimuli.

**Acknowledgments**

We are grateful to Bernard W. Balleine for sharing BD dataset with us.

## Footnotes

[1] We use the automatic differentiation in Tensorflow [Abadi et al., 2016].

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
