[Supplementary Material]

# Disentangled behavioral representations

## Supplementary Material

Amir Dezfouli, Hassan Ashtiani, Omar Ghattas, Richard Nock, Peter Dayan, Cheng Soon Ong

## S1 The model

### S1.1 Model architecture

The recurrent neural network in the encoder network consists of $N_{\text{enc}}$ bidirectional LSTM cells [Hochreiter and Schmidhuber, 1997, Schuster and Paliwal, 1997]. The final output vectors of the RNN (backward and forward) are concatenated (which will be of dimensionality $2N_{\text{enc}}$) and fed into four feed-forward fully-connected layers, with $N_{\text{enc}}$, $N_{\text{enc}}$, 10, and 2 neurons respectively. The last layer corresponds to the latent representations (in our case, with 2 neurons). The activation function for the layers are Rectified linear units [ReLU; Nair and Hinton, 2010], *ReLU*, *soft-plus* [Dugas et al., 2001] and linear respectively. The decoder network is composed of four fully-connected feed-forward layers with 100, 100, 100, 69 neurons respectively and with *tanh* activation function in all the layers. The outputs of the last layer (69 outputs) are used as the weights of the GRU-based learning network and the mapping $W^n$ from this network to the policy. The GRU network has 3 cells, which requires 63 weights, and $W^n$ has 3 (GRU outputs) $\times 2$ (actions) elements.

The learning network is small to avoid its being able to predict the input signal without relying on the latent representations, a phenomenon known as *posterior collapse* in variational autoencoders [e.g., Bowman et al., 2015, Kingma et al., 2016]. To assess the extent of posterior collapse, we calculated a *random* reconstruction error during training by assessing how well a network generated based on the latent representation of one sequence could predict the actions of another. If the learning network is weak enough that it cannot ignore the latent representation, then its performance should be compromised by this shuffling. Figures S3 and S7 show that the random reconstruction loss is indeed higher than the training loss for both the SYNTHETIC and BD data, which indicates that posterior collapse has been successfully inhibited.

### S1.2 $\mathcal{L}_{\text{DIS}}$ term

$\mathcal{L}_{\text{DIS}}$ is composed of two components: an MMD term and a KL term,

$$\mathcal{L}_{\text{DIS}} = \lambda_1 \text{MMD}(\hat{q}(\mathbf{z}), p(\mathbf{z})) + \text{KL}(g(\mathbf{z}) \| p(\mathbf{z})). \tag{12}$$

For the calculation of the MMD term, a Gaussian Radial basis function (RBF) kernel was used to turn the samples $\mathbf{z}^n$ into a smooth distribution. The kernel hyperparameters that maximized the MMD were used. [2]

In principle, the MMD term should suffice to force the latent distribution to follow the prior distribution $p(\mathbf{z})$. However, there are other measures of discrepancy between distributions, and the possibility of combining various such. We found it best to include contributions from the KL divergence between $p(\mathbf{z})$ and a Gaussian match ($g(\mathbf{z})$) of the mean ($\mathbf{m_z}$) and covariance ($\text{Cov}_{\mathbf{z}}$) of $\hat{q}(\mathbf{z})$.

Theoretically, combining the two losses in fact makes sense from a geometric standpoint. Under some assumptions, the MMD term can be related to integral probability metrics and then Kantorovich-Rubinstein formula [Villani, 2009, Particular case 5.16]: it is thus an optimal transport dissimilarity between the supports of $\hat{q}$ and $p$ (the data manifold). On the other hand, since $g$ and $p$ are Gaussian, it is also a known result that the KL divergence between two Gaussians equals a sum of the LogDet divergence between their covariance matrices [Kulis et al., 2009] and a Mahalanobis divergence

between their averages [Amari., 1985]: it thus defines an information geometric distortion between their parameters [Amari and Nagaoka, 2000] (the statistical manifold).

## S1.3 Training the model

$\mathcal{L}_{\text{SEP}}$ can itself induce a local optimum form of collapse associated with ignoring one of the latent dimensions. If, for instance, $|\partial u_t^n / \partial z_1|$ is uniformly small, then $\left| \frac{\partial^2 u_t^n}{\partial z_1 \partial z_2} \right|$, and thus $\mathcal{L}_{\text{SEP}}$ will be small. This can persist as the gradients of the weights that pass through the ignored latent dimension will continue to remain small throughout the training process and therefore the weights will not get updated.

Thus, we trained the model in stages. In stage 1, we started by setting $\lambda_3 = 0$ (eliminating $\mathcal{L}_{\text{SEP}}$) and chose the optimal number of training iterations using cross-validation (early-stopping). Then, in stage 2, we introduced $\mathcal{L}_{\text{SEP}}$ term using a small $\lambda_3$ ($\lambda_3 \leq 1$), whist ensuring that the reconstruction loss was not materially affected, thus avoiding collapse.

All the gradients were calculated using automatic differentiation and the models were optimised using Adam optimiser [Kingma and Ba, 2014]. In the stage 1 of training, all the parameters ($\Theta_{\text{enc}}$ and $\Theta_{\text{dec}}$) were optimised together. In the stage 2 of training (in which $\mathcal{L}_{\text{SEP}}$ was introduced), $\Theta_{\text{enc}}$ and $\Theta_{\text{dec}}$ were optimised iteratively. That is, $\Theta_{\text{dec}}$ was optimised for 50 times and then $\Theta_{\text{enc}}$ was optimised for 200 times, and then this loop was iterated.

Since some subjects might have a bias towards one of the actions ($C_1$ or $C_2$), we randomly counter-balanced $C_1$ and $C_2$ between the encoder and decoder in order to prevent the latent representations from being affected by such biases.

We used an Accelerator Cluster for running the experiments (with NVidia Tesla P100 (SXM2)).

## S1.4 Model parameters

For BD data we set $N_{\text{enc}} = 20$. For SYNTHETIC data, we observed that $N_{\text{enc}} = 20$ did not fit the behaviour adequately (i.e., reconstruction error was still decreasing after >60000 training iterations without increasing the test reconstruction error), and therefore we used a more powerful encoder by choosing $N_{\text{enc}} = 50$ for SYNTHETIC experiment.

In stage 1 of the optimisation (i.e., loss without $\mathcal{L}_{\text{SEP}}$ term – see above) the hyper-parameters were $\lambda_1 = 50$, $\lambda_2 = 1$ and $\lambda_3 = 0$. For the second stage of the optimisation the hyper-parameters were $\lambda_1 = 50$, $\lambda_2 = 1$, $\lambda_3 = 1$ for SYNTHETIC experiment. For BD experiment we found that $\lambda_3 = 1$ at stage 2 of training causes an increase in the reconstruction loss during optimisation, which was then avoided by decreasing $\lambda_3$ to 0.1. The other two parameters were also decreased to $\lambda_1 = 2$, $\lambda_2 = 0.5$. In this settings the $\mathcal{L}_{\text{SEP}}$ decreased without materially increasing $\mathcal{L}_{\text{RE}}$ during stage 2 optimisation.

## S1.5 On-policy simulations

For the on-policy simulations, the actions were selected by model based on the probabilities predicted by the model. The first action that was fed to the model was $C_1$ and the remaining actions were selected by the model.

## S2 SYNTHETIC data

For generating synthetic data, the agents learned and selected actions as follows. At each time step $t$, after agent $n$ selected action $a_t^n$ and received reward $r_t^n$, the $Q$-value of action $a_t^n$ was updated according to $Q_{t+1}(a_t^n) = (1 - \alpha)Q_t(a_t^n) + \alpha r_t^n$. $\alpha$ is the learning rate, and was fixed to 0.2 for all the agents. The action in the next trial was selected according to the following probability,

$$p(a_{t+1}^n = C_1) \propto e^{\beta^n Q_{t+1}(C_1) + \kappa^n \mathbb{I}[a_t^n = C_1]}, \quad p(a_{t+1}^n = C_2) \propto e^{\beta^n Q_{t+1}(C_2) + \kappa^n \mathbb{I}[a_t^n = C_2]}, \quad (13)$$

in which $\mathbb{I}[.]$ is the indicator function.

The perseveration parameter for agent $n$ was drawn randomly from a Gaussian distribution,

$$\kappa^n \sim \mathcal{N}(0, 1). \quad (14)$$

The inverse temperature parameter was selected randomly according to,

$$\epsilon^n \sim \mathcal{N}(0, 36), \quad \beta^n = |\epsilon^n|. \tag{15}$$

We generated 1500 agents, each of which selected 150 actions. 30% of agents were used for testing. The actions paid off probabilistically as either $\{p(r = 1|a = C_1) = 0.1, p(r = 1|a = C_2) = 0.5\}$, or $\{p(r = 1|a = C_2) = 0.1, p(r = 1|a = C_1) = 0.5\}$, counterbalanced randomly across the agents.

Since the latent space has only two dimensions, we were able to directly visualize/report one-to-one relationships between each latent variable and each factor of variation in the data (Figures 2,S1). We also calculated the disentanglement metrics reported in disentanglement_lib [Locatello et al., 2018] based on the results in the synthetic dataset obtained by including the separation loss, and without including the separation loss, which showed that the separation loss improved disentanglement.

## S3   BD dataset

In this dataset, each subject completed the task 12 times (12 blocks), so each generated 12 input sequences. Of those sequences, a randomly selected 8 were used for training the model and the remaining 4 for testing. On average, participants completed 109.45, 114.91, 102.79 trials per block in healthy, depression, and bipolar groups respectively.

To generate Figure 4(a), we calculated the mean pairwise distance between the latent representations of each subject $i$ ($d_{\text{within}}^i$), and the pairwise distance between the latent representations of each subject $i$ to the other subjects ($d_{\text{between}}^i$). For calculating $d_{\text{within}}^i$, note that there are 12 latent representations for each subject. For each of these representations, we calculated the total of 66 non-trivial, unique, distances arising from the representation of input sequence $i$ to that of the input sequence $j$ ($\forall j > i$). Denote by $\mathbf{z}^{ik}$ the latent representation for the $k$th input sequence of subject $i$,

$$d_{\text{within}}^i = \frac{1}{66} \sum_{k=1}^{12} \sum_{l=k+1}^{12} \|\mathbf{z}^{ik} - \mathbf{z}^{il}\|_2. \tag{16}$$

For calculating $d_{\text{between}}^i$, we calculated the distance between latent representations of subject $i$ and each subject $j$ ($i \neq j$),

$$d_{\text{between}}^i = \frac{1}{(66 + 12)(N - 1)} \sum_{j=1, j \neq i}^{N} \sum_{k=1}^{12} \sum_{l=k}^{12} \|\mathbf{z}^{ik} - \mathbf{z}^{jl}\|_2. \tag{17}$$

Note that there are $66 + 12$ pair of distances between each two subjects, as the distance is symmetric.

## S4   Generalized separation loss

In Section 3 we introduced the separation loss for the special case of only two actions and two latent variables. In this section we extend this notion to address a more general setting. Recall that $\pi_t^n(.)$ was the vector of probabilities that the learning network assigns to the actions (at trial $t$ and for the subject $n$). Let $\pi_t^n(C_{\text{max}})$ be the maximum value of these action probabilities, and let $\pi_t^n(C_{\text{alt}})$ be the second highest of these action probability. Define

$$u_t^n = \log\left(\frac{\pi_t^n(C_{\text{max}})}{\pi_t^n(C_{\text{alt}})}\right), \tag{18}$$

which basically measures the learning network's confidence in its preferred action (compared to the "alternative" second best action). Note that Eq. 18 is consistent with the definition that we had for $u_t^n$ in Section 3 since for the special case of two actions we have

$$u_t^n = \log\left(\frac{\pi_t^n(C_{\text{max}})}{\pi_t^n(C_{\text{alt}})}\right) = \log\left(\frac{\frac{e^{v_t^n(C_{\text{max}})}}{\sum_i e^{v_t^n(C_i)}}}{\frac{e^{v_t^n(C_{\text{alt}})}}{\sum_i e^{v_t^n(C_i)}}}\right) = \log\left(\frac{e^{v_t^n(C_{\text{max}})}}{e^{v_t^n(C_{\text{alt}})}}\right) = v_t^n(C_{\text{max}}) - v_t^n(C_{\text{alt}}), \tag{19}$$

where $C_{\text{max}}$ and $C_{\text{alt}}$ are the actions corresponding to $\pi_t^n(\text{max})$ and $\pi_t^n(\text{alt})$ respectively. Ideally, the effect of different $z_i$'s on changing the behaviour (i.e., the preferred action) should be independent of each other. The following notion, which is analogous to Eq. 8, captures the pairwise interactions between $z_i$ and $z_j$ on changing the preferred action

$$d_t^n(i,j) = \left| \frac{\partial^2 u_t^n}{\partial z_i \partial z_j} \right| . \tag{20}$$

In order to be able to interpret each latent variable independently, we minimize these pairwise interactions over all the choices of $z_i$ and $z_j$. In particular, we aim at minimizing the following loss function

$$\hat{\mathcal{L}}_{\text{SEP}} = \frac{1}{M(M-1)N} \sum_{n=1}^{N} \sum_{t=1}^{T^n} \sum_{\substack{z_i, z_j \\ z_i \neq z_j}} \left| \frac{\partial^2 u_t^n}{\partial z_i \partial z_j} \right| . \tag{21}$$

where $M$ was the dimensionality of the latent space. Note that for the special case of $M = 2$, Eq. 21 simply recovers the definition that we already had in Eq. 9

$$\frac{1}{2N} \sum_{n=1}^{N} \sum_{t=1}^{T^n} \sum_{\substack{z_i, z_j \\ z_i \neq z_j}} \left| \frac{\partial^2 u_t^n}{\partial z_i \partial z_j} \right| = \frac{2}{2N} \sum_{n=1}^{N} \sum_{t=1}^{T^n} \left( \left| \frac{\partial^2 u_t^n}{\partial z_1 \partial z_2} \right| + \left| \frac{\partial^2 u_t^n}{\partial z_2 \partial z_1} \right| \right) = \frac{1}{N} \sum_{n=1}^{N} \sum_{t=1}^{T^n} \left| \frac{\partial^2 u_t^n}{\partial z_1 \partial z_2} \right| . \tag{22}$$

## Footnotes

[2] Employing the implementation: https://github.com/tensorflow/models/blob/master/research/domain _adaptation/domain_separation/losses.py

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

Figure S1: This figure is similar to Figure 2 but *without* optimising the loss function with $\mathcal{L}_{\text{SEP}}$ term. (a) Relationship between the dimensions of the latent representations ($z_1$, $z_2$) and the parameters used to generate the data ($\kappa$ and $\beta$). The black lines were calculated using method 'gam' in R [Wood, 2011] and the shaded area shows confidence intervals. (b) Off-policy simulations of the model for different values of $z_1$ (left-panel; $z_2 = 0$) and $z_2$ (right-panel; $z_1 = 0$). The plots show the probability of selecting $C_1$ in each trial when $C_1$ had actually been chosen on all the previous trials. A single reward is provided, shown by the vertical red line. (c) Model simulations similar to the ones in panel (b) but using the actual $Q-$learning model. In the left panel $\beta = 3$ and in the right panel $\kappa = 0$.

Figure S2: SYNTHETIC dataset. Off-policy simulations during model training, which shows how the effect of $z_1$ and $z_2$ on behaviour are separated after the introduction of $\mathcal{L}_{\text{SEP}}$ term. Note that iteration 0 is the beginning of the introduction of $\mathcal{L}_{\text{SEP}}$ term in the loss function. Each iteration consists of 50 updates of decoder parameters and 200 updates of encoder parameters. Note that this is only stage 2 of the training and after the introduction of $\mathcal{L}_{\text{SEP}}$ term into the loss function.

Figure S3: SYNTHETIC data. Training, test, and random reconstruction loss. Random reconstruction loss quantifies the specificity of the latent representations to each input sequence, allowing us to detect posterior collapse. To calculate this loss, the learning network for a sequence is generated based on the encoded latent representation of a radomly-selected *different* sequence. If the latent representations are specific to the input sequences we expect this random reconstruction loss to increase by training, which is the case as shown in the graph. Note that the graph only shows stage 1 of the training. See text for more description.

Figure S4: SYNTHETIC data. Distribution of $z$ values.

Figure S5: SYNTHETIC data. Off-policy simulations of the model. The top panel is similar to Figure 2(b). The bottom panel is similar to top panel, but the action fed to the model as the previous action was $C_2$ (instead of $C_1$ which was fed to the model in the top panel).

Figure S6: BD data. Off-policy simulations of the model. The top panel is similar to Figure 4(c). The bottom panel is similar to the top panel, but action $C_2$ was fed to the model as the previous action (instead of $C_1$ which was fed the model in the top panel).

Figure S7: BD data. Training, test, and random reconstruction loss. Random reconstruction loss quantifies the specificity of the latent representations to each input sequence, allowing us to detect posterior collapse. To calculate this loss, the learning network for a sequence is generated based on the encoded latent representation of a radomly-selected *different* sequence. If the latent representations are specific to the input sequences we expect this random reconstruction loss to increase by training, which is the case as shown in the graph. Note that the graph only shows stage 1 of the training. See text for more description.