[Reviews · NeurIPS 2019]

Reviewer 1



The proposed architecture consists of an encoder RNN which maps input sequences to low-dimensional representations, a decoder network which maps this latent representation to weights of an RNN (similar to Hypernetwork). This RNN is used to predict labels in the input sequence. The whole network is trained end-to-end. This sort of an RNN-based auto-encoder + hypernetwork combination is novel and interesting. Results on a synthetic time-series dataset and the BD dataset show that the proposed architecture is indeed able to disentangle and capture factors of the data generating distribution. Experiments overall, however, are quite limited. It would be great to exhaustively compare to 1) an RNN-based autoencoder without the hypernetwork, 2) ablative experiments with and without the disentanglement and separation losses, 3) within the disentanglement loss, contribution of MMD vs. KL and analysis of the kinds of differences / behavior each induces. This would help tease apart the significance of the proposed architecture. Also, disentanglement_lib (https://github.com/google-research/disentanglement_lib) is a recent large-scale benchmark and presents several standard metrics for evaluating disentangled representations. It would make for a significantly stronger contribution to evaluate on this and compare against prior work. As things currently stand, the proposed architecture is interesting but hasn't been evaluated against any prior work.

Reviewer 2



After rebuttal: I focused my comments and attention on the utility of this method on providing behavioral representations, whereas R3 and the authors drew my attention to the novelty of the separation loss, and their specific intent to primarily model _individual decision-making processes_, and not behavior more generally. The text could use clarification on this point in several places. I still do think that existing PGM-based approaches with subject-level random variables are a fair baseline to compare against, since they do create latent embeddings of behavior on a per-subject basis (with e.g. per-subject transition matrices in HDP-HMM models of behavior), but want to recognize the novelty of the architecture and approach. -------------------------------------------------- Originality: There is a large body of work for machine learning modeling of time-series behavior with interpretable latent spaces. I find the originality low in the context of that prior work, including Emily Fox's work on speaker diarization and behavioral modeling, Matthew Johnson's recent work on structured latent space variational autoencoders, and others. Given the input of the model is a bag of low-dimensional sequences, probabilistic graphical models are an appropriate baseline here. Quality: The writing, construction of the model, and evaluation of the model are sound. However, the model is not compared to any alternatives. It is difficult for me to place an absolute significance on the work if it is not compared to even a naive or strawman baseline. For instance, if you chunked the input sequences, did PCA on all chunks, and averaged the embedding, how close would that get you to the properties of the model being proposed? Is an RNN even necessary? If the ultimate goal (or at least stringent evaluation) of this model is to tell apart different treatment groups, then set up a classification task. Clarity: The work is clearly written and the figures are well-constructed and present information clearly. Significance: The work is if low to moderate significance, given the context the work is in. Directly comparing to alternate ways of capturing an interpretable latent space would help lend significance, if this method was indeed better, faster or easier to use than alternate methods. The model is clearly capturing task-related information in figure 4, but I have no idea if an RNN is required to do this, or if a much simpler method could do the same. Without this backstop, I don't know if this is an interesting method.

Reviewer 3



After rebuttal: Having read the rebuttal and other reviewers' comments, I still think this is a strong paper. I don’t consider the novel contribution to be the disentanglement of representations, but rather the separation loss and interpretability, which they have done appropriate ablations for. A more thorough set of ablations on all of the different components would have been nice to see, but unnecessary in my mind, since they don’t claim their main contribution to be hypernets or disentangling. Furthermore, they’ve validated (to a limited extent) on a real-world dataset, which was key to my high rating. That said, they could have gone much further in this respect, including addressing my point about needing to know the exact number of latent dimensions and how not knowing this would affect their method. I encourage the authors to consider adding this if accepted. R2 does have a point about comparing to some non-RNN baselines, which would have made the paper stronger. The Dezfouli et al 2018 paper was using only supervised learning, and doesn’t consider other methods suggested by R2 like HMM (although it didn’t have the same goals as the current paper). ======================================== Before rebuttal: This paper introduces a new method of training a deep encoder-recurrent neural network in order to map behavior of subjects into low-dimensional, interpretable latent space. Behavioral sequences are encoded by an RNN to map into a latent space, which are then decoded and used as the weights (hypernet style) of a second RNN, which is trained to predict the behavioral sequences of each subject. The training loss includes a reconstruction loss, a disentanglement loss, and a separation loss. The separation loss is a novel contribution, and encourages the effect of the latents on behavior to be separable. The analyses and figures are extremely well-done and clear. They show that on a synthetic dataset, generated by a Q-learning agent with 2 parameters, this method can recover latents that correspond to these 2 parameters in an interpretable way. Without the separation loss, this doesn’t occur. They further tested on a real-world dataset consisting of behavioral data from subjects with depression and bipolar disorder and healthy subjects. They found that one of the latent dimensions could differentiate between the groups, and further that the distances in latent representations between groups was larger than within groups, validating the usefulness of their approach. The explanations were clear and at an appropriate level of detail. The figures are great, I especially found the off-policy and on-policy simulations to be illuminating and very nice. Along with the supplementary text, the experiments seemed quite thorough. However, I would have liked to see what happens when the number of latent dimensions doesn’t exactly match the number of parameters in the agent generating the synthetic dataset. For real behavioral data, we don’t know the number of relevant dimensions, and it’s not clear to what extent this method relies on knowing this. The separation loss is quite novel. To what extent can it be generalized to considering not only separating choices, but also options? Could the authors provide an intuition for how good the approximation in equation 10 is? Very minor: please provide a citation for “... which is consistent with the previous report indicating an oscillatory behavioral characteristic for this group”.

[Author Response · NeurIPS 2019]

We thank the reviewers for their comments.

**Reviewer #1:▷ Role of each term**.

> We actually showed the unique effect of the separation loss over the other terms in suppl. Fig. S1 and S2.

Fig. S1 shows the results without the separation loss; Fig. S2 shows how the results change after the introduction of separation loss during training. We, therefore, respectfully disagree with the reviewer's comment that "it is unclear how much 3 contributes over 2". This is also pointed out in the main text in lines 177 and 178. We could move this material to the main text in light of the reviewer's comments. The other two terms in the loss function are similar to the standard terms taken from Tolstikhin et al, 2017 cited in the paper, which is why we didn't establish their usefulness here (the KL term experimentally improves optimization, but we do not consider it as our contribution here). Further, RNNs without hyper-networks are studied elsewhere (Dezfouli et al 2018a) in the same BD dataset that we used here, so their relative performance is known (they showed important aspects of decision-making remained uncaptured by typical computational models and even their enhanced variants, but were captured by RNNs automatically).

▷ **Comparison with other works**. Without the separation loss, the autoencoder framework is equivalent to Tolstikhin et al, 2017 (without RNN and hyper-net), which as we showed (Fig. S1; S2) does not disentangle effectively. Therefore we are indeed comparing our framework with this previous work.

Please also note that since the latent space has only two dimensions, we were able to directly visualize/report one-to-one relationships between each latent variable and each factor of variation in the data (Fig 2a, S1a). Following the reviewer's comment, we calculated the disentanglement metrics reported in disentanglement_lib (Locatello et al, 2019) based on the results in the synthetic dataset obtained by including the separation loss, and without including the separation loss. In all the metrics the separation loss improved disentanglement (with separation loss > without separation loss): MIG: 0.29>0.11, DCI: 0.19>0.03, SAP: 0.15>0.06, $\beta$-VAE score: 1>0.99, factor-VAE score: 0.94>0.68, modularity: 0.99>0.87. Please refer to Locatello et al, 2019 for the meaning of each metric.

▷ **Quadratic cost w.r.t number of latent dimensions**. The utility of our method depends on the number of latent *not* growing with the number of subjects. Current experiments are using >1000 sequences, which is considered to be on the high-end of the number of subjects in psychological/neuroscience studies.

**Reviewer #2:▷ Limited novelty compared to previous works**.

> Our aim is NOT to represent or classify behavioural trajectories as a generic time series (#1, for short), but to characterize differences in the (typically causal) processes underlying reinforced choice (#2).

These aims are very different. From the references cited, the reviewer might be under the misapprehension that we are solving #1. For instance, Johnson et al 2016 build an interpretable representation of movements of a mouse in an experiment (# 1). This framework is NOT able to extract how the individual differences in such movements can be explained in a low dimensional space (#2). The same applies to Fox el al, 2011. Indeed, these two frameworks are functionally equivalent to the learning network in the current architecture. As such, we believe that the aims and the architecture of the current framework are quite different from the previous models on disentanglement on general time-series. We can, of course, discuss theses references in the paper and explain their differences with our model.

▷ **Usefulness of RNNs for modelling human learning processes**. On the encoder side, the reviewer suggests using PCA instead of an RNN to extract the features of the learning processes. Even on the very same dataset (BD), it is shown that both linear models and more complex cognitive characterizations fail to learn the complex patterns in human behaviour (Dezfouli et al 2018a; cited in the paper, which has now been published in PLOS Computational Biology). Given the constraints of disentanglement, we fail to see the merit of employing a weaker technique.

▷ **Ultimate aim is classification**. We are working in an unsupervised setting with the aim of characterizing individual differences. Psychiatric classification is notoriously crude; we just used them in the BD dataset for coarse validation.

▷ **Comparison with previous work**.

> We do show that the previous autoencoder architecture (Tolstikhin et al 2017) fails to produce desirable disentanglement results here and our new separation loss is required. Please see Figure S1 and S2.

**Reviewer #3**: Apologies for the short response.

▷ **Separation loss for options**. In principle the separation loss can be used over the space of options instead of actions, which could be interesting since it allows analyzing high-level strategies.

▷ **Intuition behind equation 10**. The tightness of equation 10 is intuitively related to whether the direction of the effect of $z_1$ and $z_2$ on behaviour depends on $t$. For example, in the simulations here changing $z_1$, $z_2$ affects action probabilities in the same direction (increase or decrease) across different time steps, which makes the approximation more accurate. We plan to derive more formal results about this approximation in future works.

[Meta-Review · NeurIPS 2019]

This paper does provides a method for fitting a recurrent network to human behavioural data in an interpretable way. This is achieved via kind of auto-encoding process, where behaviors are encoded into a latent space, which are then decoded into the weights of a second RNN, which is trained to predict subject-specific data. The three expert reviewers all recommended this work for acceptance, although they pointed out some important limitations. R1 in particular was concerned about the lack of ablations; it is not immediately clear to the reader how the quantify the importance of various components of the proposed approach (such as hypernets). R2 and R1 were also concerned about the lack of comparison with any established method, and R2 was keen to see a comparison with the more immediately transparent 'cognitive model' approaches that the authors hint at in the abstract of the paper. Despite these limitations, the overall consensus was that the work is of high quality and makes important contributions to how RNNs can be applied to understand human behaviour and decision making. As such, I think it warrants acceptance to the conference.